# Influence of Materials Parameters of the Coil Sheet on the Formation of Defects during the Manufacture of Deep-Drawn Cups

**Wojciech Baran [1],\*, Krzysztof Regulski [2] and Andrij Milenin [2]**

[1] Can-Pack S.A., Business Support Service, 32-800 Brzesko, Poland

[2] Faculty of Metals Engineering and Industrial Computer Science, AGH University of Science and Technology, 30-059 Kraków, Poland; regulski@agh.edu.pl (K.R.); milenin@agh.edu.pl (A.M.)

\* Correspondence: wojciech.baran@canpack.com; Tel.: +48-796-737-055

**Abstract:** During the process of deep drawing of cylindrical thin-walled products from aluminum sheets, the occurrence of product defects in the form of breaking the material continuity is observed. This has a very large impact on the efficiency of production lines and the number of generated scraps. The number of defects depends on many factors, including the material and the process properties. Because the problem appears after changing one material to another, while the process parameters do not change, it was assumed that the material has the main influence on the number of defects. To reduce the number of defects, a tool is needed to predict threats to the process. Decision tree models were used for this purpose. Using the tree interaction algorithms, the influence of the chemical composition and strength parameters of the 3xxx series aluminum alloy on the number of generated defects was investigated. Increased Silicon (Si) and Iron (Fe) values generated a higher number of defects. Increased yield strength (YS) and decreased elongation (E) also generated a higher number of defects. Based on the results, a defect prediction tool was created, where after entering the parameters of the material, it is possible to predict production hazards.

**Keywords:** short cans; damaged cans; mechanical parameters; chemical composition; influence; correlations; decision tree models; regression tree; classification tree; model C&RT

## 1. Introduction

Manufacturing enterprises place a great emphasis on the optimization of the production process. They can achieve it by, among other things, increasing the efficiency of the production lines. One of the activities that has a significant impact on the improvement of the production efficiency is decreasing the number of defects in products, which are generated during the forming process. When a beverage can manufacturer produces 18 billions cans per year [1], decreasing spoilage by one percent results in 180 millions of non-wasted products. That is why in this type of business, reducing defects is so important. Since many factors influence the formation of defects, it is difficult to clearly determine which of them is significant or what combination of them gives rise to the greatest number of defective products. In the production process of an aluminum beverage can, there are potentially several defects that can arise at various stages of the process. Many of the defects can be generated on the front of the line, on the vertical press during cup operation, and on a horizontal press during redrawing and drawing operations as well as forming of the bottom of the can. One of the most common defects on the horizontal press is related to the loss of material continuity when aluminum is drawn between tools, which are successively reducing the can's diameter and which are called "ironings". The phenomenon of rupture of the can wall is called "short can", which is related to the lower height compared to a full-value product shown in Figure 1. Figure 1b shows a properly manufactured can. The top of the can is uneven, which is the normal effect of the material being pulled through

the ironings. On the Figure 1a it can be seen that the can is significantly shorter than the can shown in Figure 1b, which is caused by the breakage and detachment of the rest of the material during the ironing process. More details about the production process of the aluminum can are described in Section 2.2—Technology.

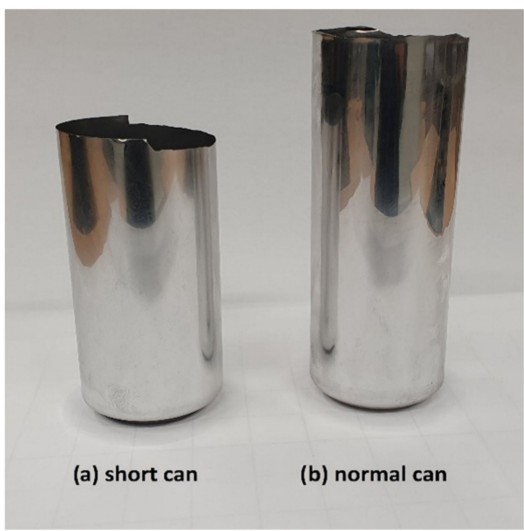

**Figure 1.** Analyzed defect—"short can".

In the literature on the production processes of aluminum beverage cans, we can find information on the influence of individual process parameters on the pressing force. Many analysts in this field tried to find such dependencies using various methods that would indicate a solution to eliminate the problem of a defective semi-finished product/final product.

Der-Form Chang, Jyhwen E. Wang (1997) [2] investigated the influence of tool angles reducing the thickness of the side wall (ironings), the influence of ironing diameter reduction and the influence of friction on the deep-drawing process. They extended the analysis of deep-drawing parameters taking into consideration the material's inhomogeneous deformation. They proved that punch load increases with increasing thickness reduction, die-cup coefficient of friction, punch-cup coefficient of friction, and strain-hardening coefficient and with a decreasing die semi-angle.

Similarly, Luis Fernando Folle et al. (2008) [3] dealt with the influence of the friction coefficient, ironing angles, but in terms of the influence on the pressing force. Moreover, they analyzed the influence of material hardening and the clearance between the punch and the ironing on the pressing force. They observed that the friction coefficient has the largest impact on the ironing force, the clearance between a punch and an ironing die also has a high influence on it, but smaller than the friction coefficient. Ironing die angle and strain-hardening exponent have a lower impact on the ironing force.

Marco Schunemann et al. (1996) [4] extended their research to the fundamental but lesser-known aspects of the process, such as the temperature generated during the drawing process. They wrote about the influence of deformation on temperature and lubrication. They showed that higher flow stresses in the upper part of the cup wall result in higher temperatures in that area, compared with lower cup parts and the bottom. They also showed the temperature distribution in the deformation area for the punch, material, and ironing die.

V.M. Simões et al. (2013) [5] studied the influence of lubrication on the coefficient of the friction between the material and the tools using an experimental device. They used AA5xxx series aluminum for testing. This has been supported by numerical simulations using the finite element method. Experimental results indicated that the amount of the lubricant has a negligible effect on the punch force. These results indicate that maybe

lower amounts of lubricant should be tested to better understand the effect of this variable. Similar results were obtained in the experimental and numerical analysis for the different contact conditions between the sheet and the tools. However, the contact between the sheet and the die has the highest influence on the process.

G. Venkateswarlu (2010) et al. [6] analyzed the influence of the blank temperature, ironing angle, and punch speed on the characteristics of AA7xxx series aluminum using the finite element method. Their paper illustrates the use of FE simulations with Taguchi's design of the experimental technique to determine the proportion of contribution of the important process parameters on the deep-drawing process. The blank temperature (84.4%) has a major influence on the deep-drawing process, followed by the punch velocity (9%) and die arc radius (6.6%).

GAO En-zhi (2009) at al. [7] researched the influences of material parameters such as hardening exponent $n$, yield stress $\sigma_s$, and elastic modulus $E$ on the process by a 3D finite element model simulation. They developed a model of a thin-walled hemispheric surface part. The results show that when $E$ increases but $\sigma_s$ or n decreases, the equivalent plastic strain increases, and generally, the maximum equivalent plastic strain occurs at a wall region outside the die corner. However, when the value of n decreases to 0.03 or $\sigma_s$ is smaller than 120 MPa, a higher equivalent plastic strain occurs at ball top. When n, E or $\sigma_s$ increases, a higher punch force occurs, and the influences of $n$ and $\sigma_s$ on the punch force are more notable.

The influence of the yield stress model on the forming of the material during the beverage can manufacturing process was studied theoretically and experimentally by Wędrychowicz (2021) et al. [8]. In that work, it is shown that when the stress–strain curve changes by several percent, the result of the FEM modeling of the process changes significantly. At the same time, the refinement of the dependence of the stress on strain enabled describing the features of the forming observed in experiments using the FEM simulation. Thus, it has been established that the sensitivity of the forming of the can billet to the yield stress model is quite high. For this reason, a variation in this parameter in the billets of different manufacturers can lead to the appearance of defects in the material during the forming processes.

A. Rękas (2015) at al. [9] analyzed the influence deformation range, plasticity margin, and strain-hardening factor on the efficiency of the process. They also checked the influence of the number of defects on the amount of worn tools. They proved that the increased plasticity margin (yield strength divided by ultimate tensile strength, YS/UTS) and strain-hardening factor causes an increase of the spoilage. Another conclusion was that decreasing the elongation increases the amount of damaged products. Additionally, in another article [10], they compared mechanical properties of aluminum series 3XXX from six coil suppliers to find the differences and the best coil manufacturer. Similar topics were raised also by other authors [11,12]; however, the above studies do not contain a practical analysis of what parameters of the workpiece, and to what extent, affect the formation defects in a real technological process.

Due to the difficulty of deterministic prediction and unambiguous determination of the impact of all parameters on the product, the statistical method was used for the analysis. This method focuses on observing the actual production, gathering information on the number of horizontal press jams caused by material loss, and finding relationships between material parameters and the number of defective products. Basing our study on statistical calculations will allow for considering the influence of input parameters on the result, taking into consideration all the phenomena occurring during the can manufacturing process. The test determines the influence of the chemical composition and basic mechanical parameters of the material on the number of defects.

The aim of the study is to present which of the physicochemical parameters have the greatest impact on the occurrence of the "short can" defect in question and which ranges of values contribute to increasing or reducing damage to the product.

Additionally, thanks to the use of statistical methods, on the basis of the cause-and-effect analysis, it is possible to predict what results will be generated by the material with defined parameters. The use of such a method is very helpful during production planning, at its preliminary stage, i.e., before loading the material in a coil on an uncoiler in front of the production presses.

## 2. Materials and Methods

### 2.1. Material

It is important to analyze the influence of material parameters on the number of jams. Beverage can production is mostly using aluminum alloy 3xxx series. Most of the production is performed with 3104 aluminum alloy, which is provided in H19 temper. Based on PN-EN 515:1996 standard, temper H19 strain-hardened extra hard material. Standard thickness of aluminum coils is in the range of 0.260–0.235 mm. Aluminum sheet is supplied in a rolled form called a coil. All the coils need to meet the specification based on the norm related to DWI (Draw and Wall Ironing) cans technology. The tables below (Tables 1 and 2) show the range of mechanical parameters and chemical composition for 3104 aluminum alloy for H19 tamper. There is a comparison between parameter values based on the norm and 203 coils used for research purposes. All materials used for research are in range of the norm for 3104 aluminum alloy.

**Table 1.** Mechanical parameter range for aluminum alloy 3104 and for tested material.

| Aluminum Alloy | Temper | Material Thickness (mm) | UTS (MPa) | | YS (MPa) | | Elongation $A_{50}$ (% min.) | Application | Condition Notes |
|---|---|---|---|---|---|---|---|---|---|
| | | | Min. | Max. | Min. | Max. | | | |
| EN AW-3104 EN AW-Al Mn1Mg1Cu | H19 | 0.15–0.50 | 290 | 330 | 270 | 310 | 2 | DWI cans | Cold rolled |
| Aluminum alloy 3104 coils used in research | H19 | 0.260 | 299 | 323 | 273 | 298 | 4.0–6.5 | DWI cans | Cold rolled |

**Table 2.** Chemical composition of aluminum alloy 3104 and for tested material.

| Aluminum Alloy | Si (%) | Fe (%) | Cu (%) | Mn (%) | Mg (%) | Cr (%) | Ni (%) | Zn (%) | Ti (%) |
|---|---|---|---|---|---|---|---|---|---|
| EN AW-3104 EN AW-Al Mn1Mg1Cu | 0.6 | 0.8 | 0.05–0.25 | 0.8–1.4 | 0.8–1.3 | - | - | 0.25 | 0.1 |
| Aluminum alloy 3104 coils used in research | 0.195–0.294 | 0.41–0.526 | 0.147–0.23 | 0.86–1.005 | 1.045–1.272 | - | - | - | - |

### 2.2. Technology

Aluminum beverage cans production consists of many phases, as shown in Figure 2.

At the beginning, the coil is pulled out from an uncoiler, lubricated and transferred onto the cupper press. There, from the coil, a disc is cut, held by draw pad tools and drawn into a cup. Next, on the Bodymaker machine, the cup is drawn into a can and finally cut for a proper can height. Then, the can is washed in several stages, covered by mobility chemicals and dried. Subsequently, on the decorator machine, the can is covered by lithography, external lacquer, bottom lacquer and dried in pin oven. Next, the internal surface of the can is covered using the spray machine and dried again. Finally, the can is necked, flanged, and checked by the vision system. Ready cans are packed, protected against dust and other pollutions, and prepared for transportation to brewery. Simultaneously, on the other line, the lid is prepared, which is joined with the can, after cleaning and filling, at the brewery plant.

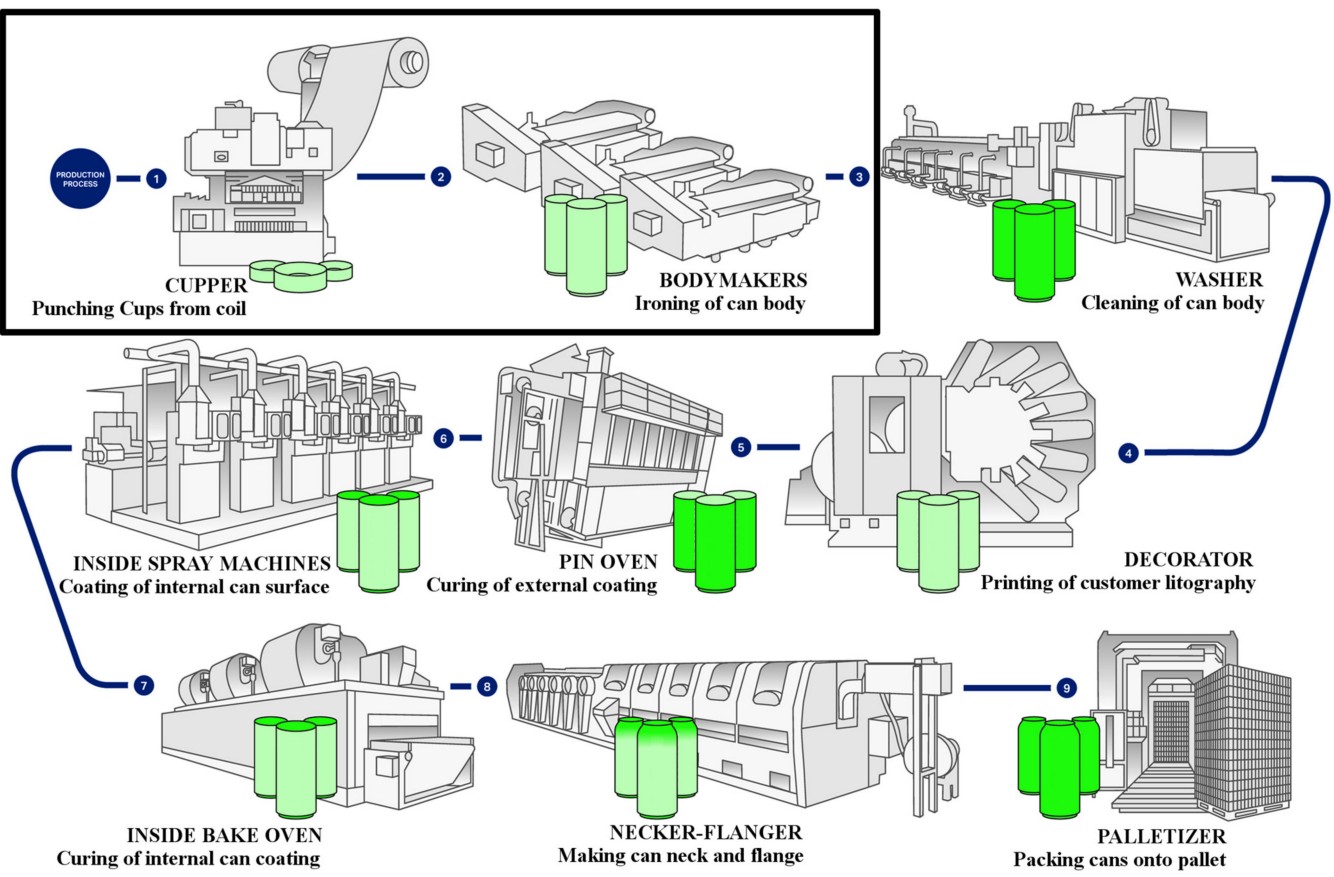

**Figure 2.** Aluminum beverage can technology process.

This article focuses on the beginning of the process, especially on the machine called the Bodymaker. In the Bodymaker, a tool pack is installed which has the task to reduce the diameter of the cup and the side wall thickness, but to extend the height of the can (Figure 3).

It happens during drawing of the cup through a set of tools such as redraw, ironing no. 1, 2, and 3 and in the last steps, the bottom of the can is formed. Each ironing has a smaller and smaller inside diameter, which causes the can wall thickness to decrease. During the reduction of the side wall thickness, sometimes, the continuity of the material is broken, generating a defect called the short can (Figure 1). The number of these defects is one of the more important factors analyzed in this article.

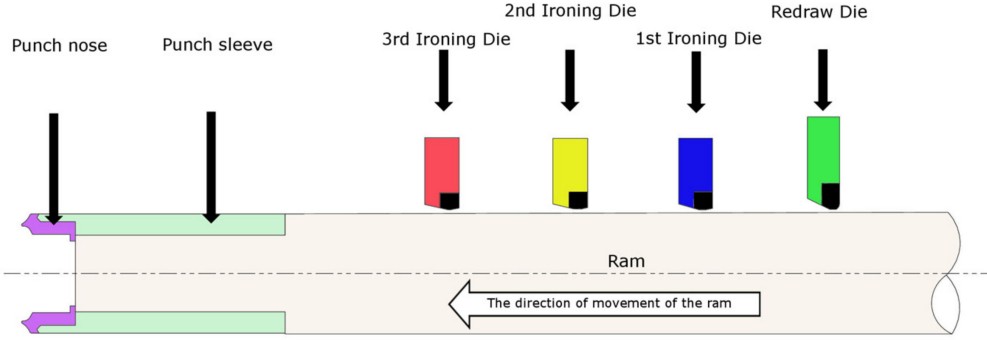

**Figure 3.** Tool pack sets installed in the Bodymaker.

*2.3. Methods*

During three months of production, for every material used for manufacturing the cans, information regarding material parameters and the amount of the short cans was collected. Coils received from manufacturers were coming with certificates, where information about chemical components content and mechanical parameters values were described. They reported the content of such chemical elements as Si, Fe, Cu, Mn, Mg, Cr, Zn, Ti, Be, B, Pb and such mechanical parameters as YS, UTS, Elongation, Ears (unevenness of the height of the side wall). Chemical composition was measured by a Spark spectrometer (SPECTROMAX-x) by using the optical emission spectrometry method. Mechanical parameters were measured by using universal tensile testing machine 50kN (MTS Exceed E43.504E) according PN-EN ISO 6892-1 norm. Tensile tests were carried out after final cold rolling in the direction of the sheet rolling.

Simultaneously, information from one of the systems for line parameters monitoring was collected. For database purposes, it was collected as data on supplier name, coil number, date and time of production from a given coil, amount of produced cans, amount of short cans, indicator (short cans/million produced cans).

All those data, both from the certificates and from the production monitoring system, were collected in one database to allow analyzing the correlation between them.

To find those correlation, we have used the STATISTICA program [13], which allows for a simultaneous comparison of a large number of data and provides a lot of tools for a deeper statistical analysis.

First, on the basis of the previously prepared data, it was determined whether there were correlations between the mechanical properties of the material and the number of jams, and between the chemical composition of the material and the number of jams. Then, it was determined whether these connections are statistically significant, and if so, to what extent. The next step was to find the range of values of these parameters that classify them into a specific group in terms of the predicted number of jams. Classification and regression tree (C&RT) was a very good tool that STATISTICA offers. An interactive regression tree was used to determine the expected number of damaged cans on the basis of the content of chemical elements and mechanical parameters. The variables (entered parameters) were discretized (divided into appropriate ranges). Then, the most important variables were selected—those that have the greatest impact on the result. The second model was developed with an interactive classification tree. This allowed—on the basis of the previously added label of sheets—to classify them into "good" and "bad", based on the number of damaged cans, to qualify sheets with the given parameters to the appropriate group.

On the basis of the generated trees (regression one and classification one), it was possible to create a set of rules that made it possible to assign a coil with the entered parameters to the range corresponding to the expected number of damaged products. Based on these rules, a tool has been created to help operators make decisions when accepting an aluminum coil for production.

## 3. Results

First, it was checked if there were differences between sheets that generate a large amount of short cans and those that do not create problems during production. For this purpose, 10 sheets were selected from all 203 coils with the largest short cans/million (SC/million) index and called bad ones. Similarly, 10 coils with the smallest SC/million index were called good ones. The mechanical parameters and chemical composition of both groups were visually compared in Figure 4. Red balls illuminate the sheet with a high SC index and blue balls show coils with a low SC index. The comparison was performed in the Paraview program. The comparison was made in the spatial coordinate system x, y, z replaced by the content of the elements Si, Fe, Mn.

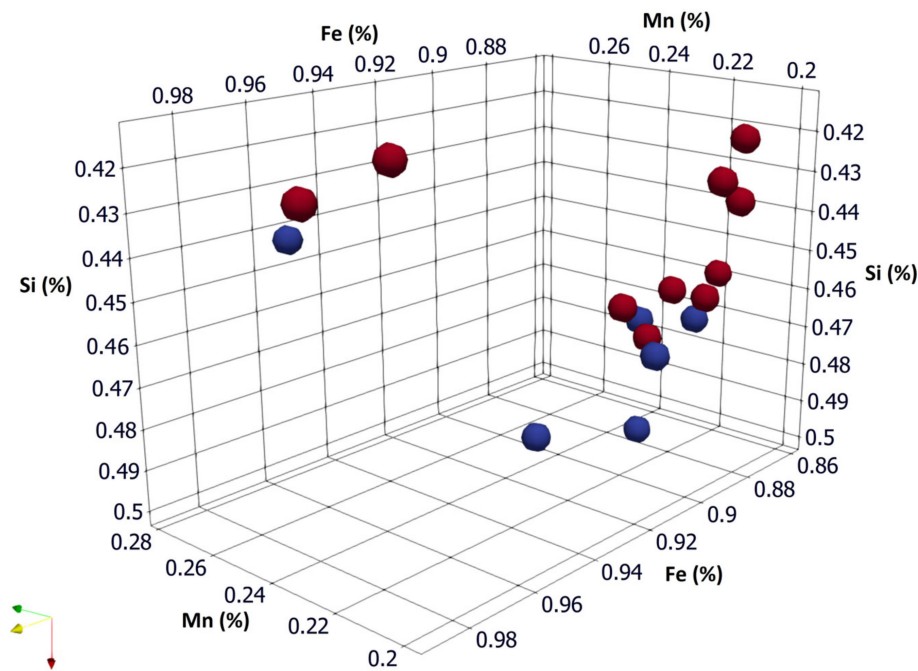

**Figure 4.** Visual comparison between coils with a high and a low short can index.

This method shows visually that we can separate "good" coils from "bad" and that increasing the Si value has an influence on generating the defects, but does not provide information about the values of all parameters and does not show their impact on the amount of defects. That is why another program was used—STATISTICA, where it was possible to draw a conclusion based on a statistical analysis. One of the most important parameters which was analyzed was the correlation between mechanical parameters and a number of short cans, shown in Table 3. Numbers written in red ($p \leq 0.05$) mean that this correlation is statistically significant.

**Table 3.** Correlation between the mechanical parameters and the index of short cans for all coils.

| Variable | Number of Short Cans/Million |
|----------|------------------------------|
| YS | 0.1664 $p = 0.018$ |
| UTS | 0.1189 $p = 0.091$ |
| Elongation | −1.767 $p = 0.12$ |
| Ears | 0.0488 $p = 0.489$ |

For visualization for Table 3, Figure 5 was prepared. The left side of the figure presents $p$ value for correlations of all mechanical parameters and the right side of the figure presents the value of the correlations. Solid blue line represents $p$ value = 0.05, which means that properties below this line are statistically significant.

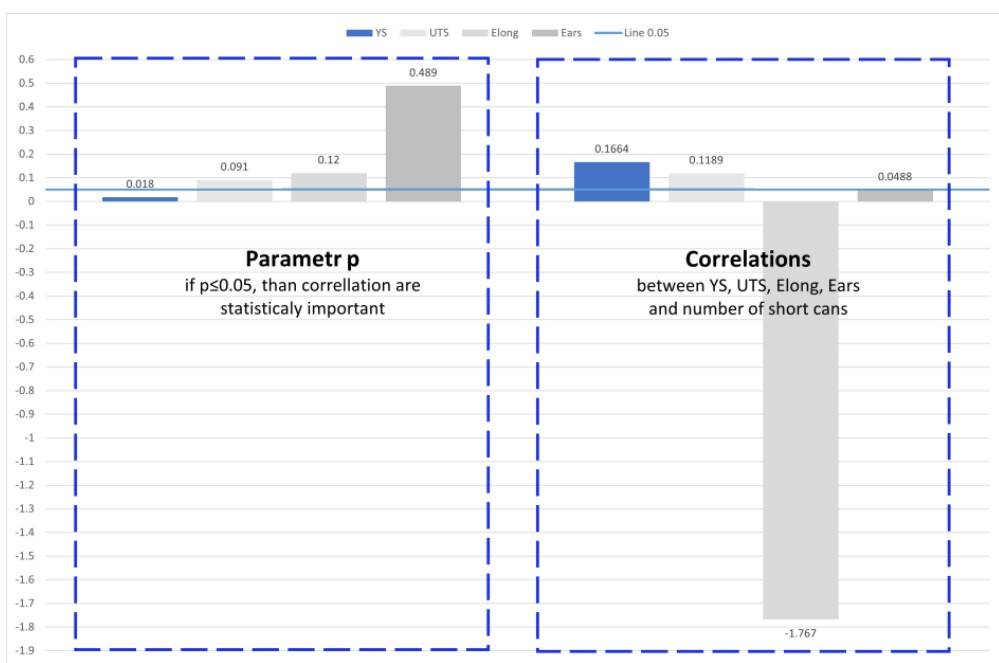

**Figure 5.** Graphical representation of correlations and *p* values of these correlations for mechanical parameters.

Table 4 shows a direct correlation between the chemical components and the index of short cans, but also an indirect correlation in the influence of chemical elements on mechanical properties, such as yield strength, ultimate tensile strength and elongation.

**Table 4.** Correlation between the chemical components and the index of short cans, YS, UTS and elongation.

| Variable | Number of Short Cans/Million | YS (MPa) | UTS (MPa) | Elongation A50 (% min.) |
|---|---|---|---|---|
| Si | 0.1771 $p = 0.012$ | 0.2262 $p = 0.001$ | 0.1357 $p = 0.053$ | −0.0258 $p = 0.715$ |
| Fe | 0.02272 $p = 0.001$ | −0.2673 $p = 0.001$ | −0.3626 $p = 0.000$ | −0.0285 $p = 0.686$ |
| Cu | 0.1212 $p = 0.085$ | −0.2852 $p = 0.000$ | −0.2634 $p = 0.000$ | 0.1822 $p = 0.009$ |
| Mn | −0.0542 $p = 0.442$ | 0.4926 $p = 0.000$ | 0.4868 $p = 0.000$ | −0.0902 $p = 0.200$ |
| Mg | −0.099 $p = 0.160$ | −0.0384 $p = 0.586$ | 0.0156 $p = 0.825$ | 0.257 $p = 0.000$ |

To better understand the above correlations, the Table 4 was made in two graphical presentations shown in Figures 6 and 7. Figure 6 shows the correlation between chemical composition and amount of short cans, while Figure 7 shows the correlation between chemical composition and mechanical parameters.

Figure 6 shows that only Silicon (Si) and Iron (Fe) have statistically important influence for generation of short cans. The other components, although having a better correlation, are not statistically significant.

Table 3 shows that Yield Strength (YS) has the biggest influence for generation of short cans. From Figure 7 it can be read, that Manganese (Mn) has the biggest influence for (YS) parameter, so it can be concluded that Yield Strength (YS) also has an indirect effect on the number of short cans.

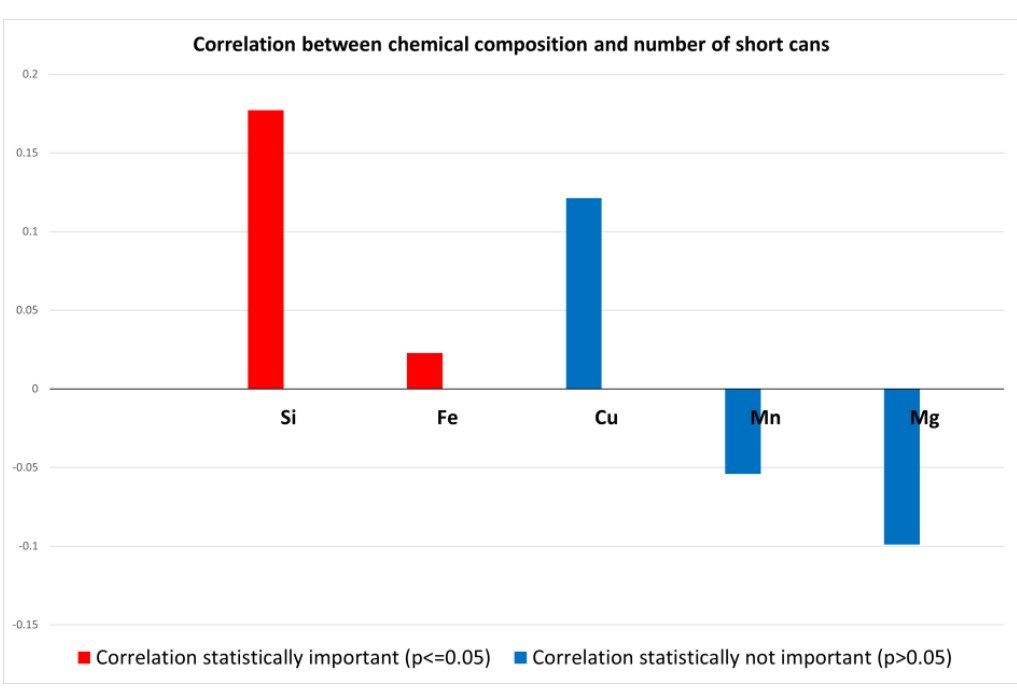

**Figure 6.** Correlation between chemical composition and number of short cans.

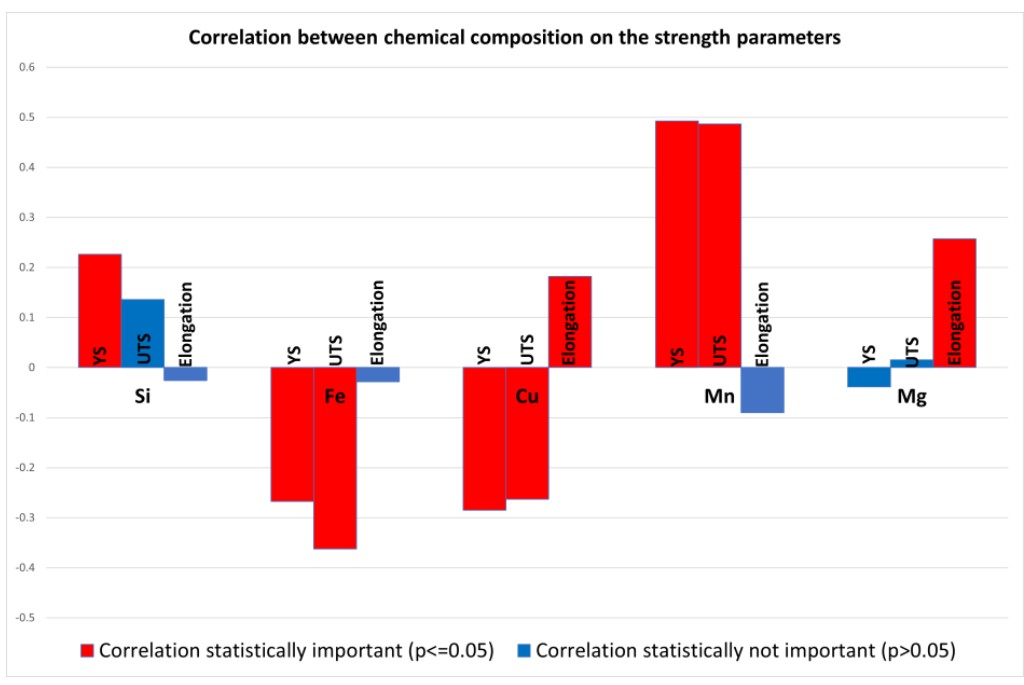

**Figure 7.** Correlation between chemical composition and strength parameters.

The determination of the correlation showed which parameters have the greatest impact on the number of defects, but on this basis, we cannot predict the impact on the number of defects for coils with different parameters. To determine this, we used an algorithm of induction of regression trees C&RT.

The C&RT algorithm was used to predict the average number of short cans. Such a tree divided the values of the number of defects into ranges, taking into consideration both the influence of the chemical composition and the mechanical parameters. Results are shown in Figure 8.

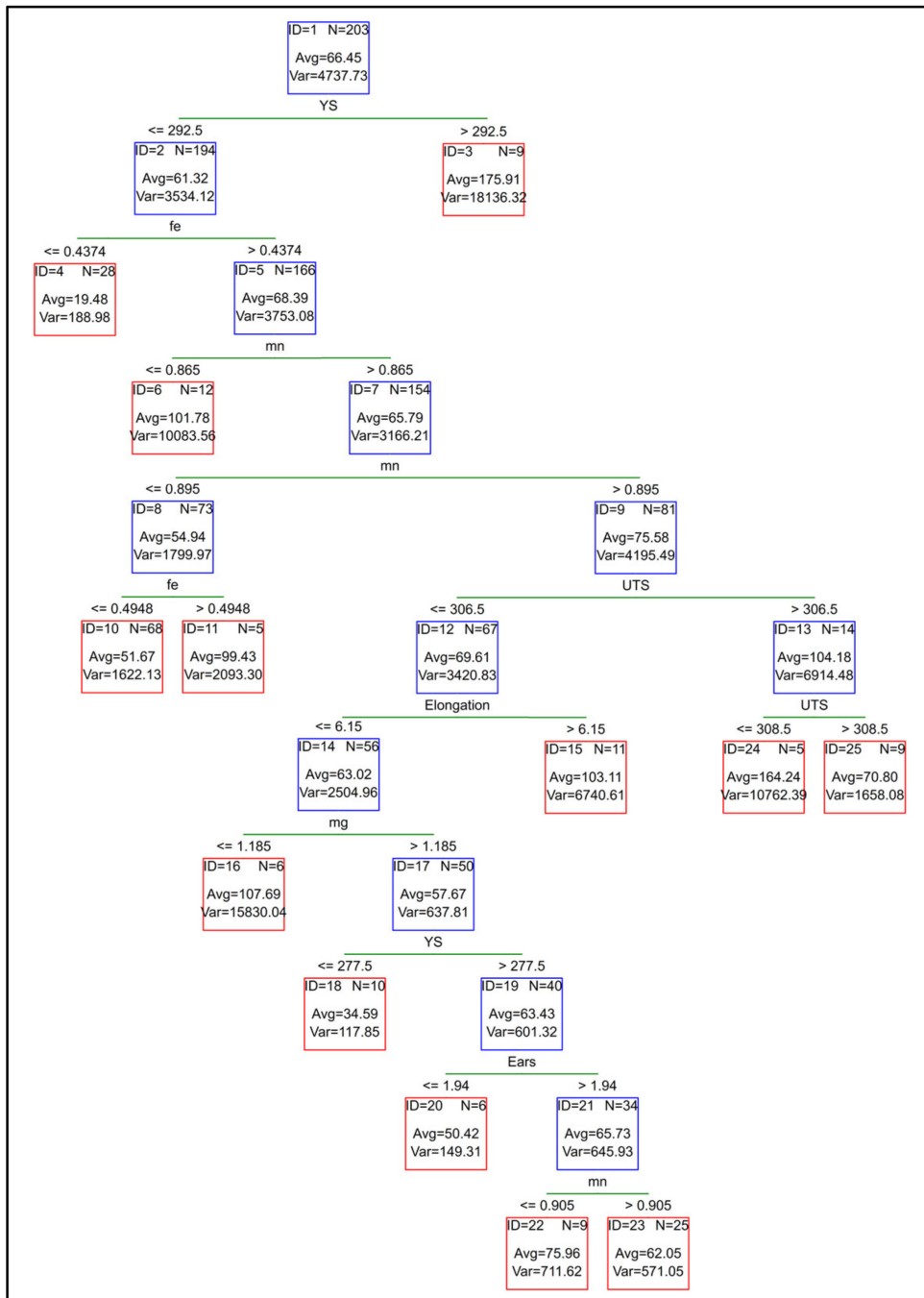

**Figure 8.** Interactive regression tree for the amount of the short cans, C&RT Model.

In general, the purpose of the tree-building analysis is to find a set of logical if-then division conditions that lead to an unambiguous classification of objects. The range of classification and regression trees (C&RT) allows for the construction of models for solving regression problems (where the dependent variable is a quantitative feature) and classification (qualitative dependent variable) [12].

The idea of C&RT trees is to split input data into ranges where the differences are the most divergent and predict the amount of damaged cans. Every leaf (every last node in Figure 8) of the tree provides one rule, which can be used for prediction.

The mean absolute error MSE (Mean squared error) for this tree is 31.75 (Table 5).

**Table 5.** Goodness of the fit of the tree model summary.

| Goodness of Fit Parameters | Parameter Values |
|---|---|
| Mean of squared residuals | 3304.97 |
| Mean squared error (MSE) | 31.75 |
| Relative mean square error (RMSE) | 0.49 |
| Relative mean deviation | 0.44 |
| Correlation coefficient | 0.55 |
| Coefficient of determination $R^2$ | 0.30 |

This error is calculated according to the formula:

$$\text{MSE} = \frac{1}{n} \sum_{i=1}^{n} \left( Y_i - \hat{Y}_i \right)^2 \tag{1}$$

where MSE—mean squared error, $n$—number of data points, $Y_i$—observed values; $\hat{Y}_i$—predicted values.

The remaining measures of fit presented in Table 5 also allow us to assess the differences between the observed values and the predicted values by the model, and these are commonly accepted measures of the model, and the method of their calculation can be found in the textbook for statistics [12].

Because the mean squared MSE error is high, the quality of the tree is not satisfying, we extended our knowledge about the dependance by building the next tree—a classification tree. All rules from both trees will be combined into one interference mechanism.

The fit of the graph (Figure 9) for low values is fairly good, but the fit for high values is divergent, thus the coefficient of determination at higher values is weak as the tree does not predict those values well. For lower values, we made predictions from this tree, but for higher values we had to use a different tool—a classification tree (Figure 10).

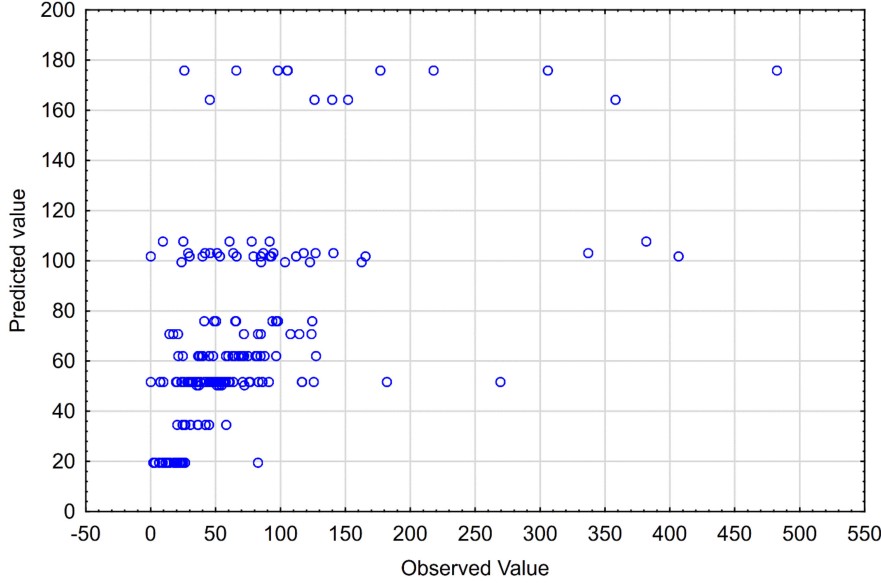

**Figure 9.** Scatter plot of predicted and observed values for the C&RT regression tree model.

In this case, all the coils were divided according to the number of short cans into BAD and NORMAL in such a way that 50% of the items with the highest number of defects (values above the median) were considered as BAD and the 50% with the lowest number of defects were considered as NORMAL. It means that 50.5 defects/million produced cans and more was considered as BAD. Less than 50.5 defects/million was considered as GOOD. This classification was called 2Q, because the amount of normal coils is two

quartile (median). This tree, like the regression tree, divides the number of short cans into areas but takes into consideration the BAD and NORMAL classifications of aluminum coils. The areas that predict the predominance of a large or a small number of defects are very visible. Moreover, we can determine the credibility and rule support.

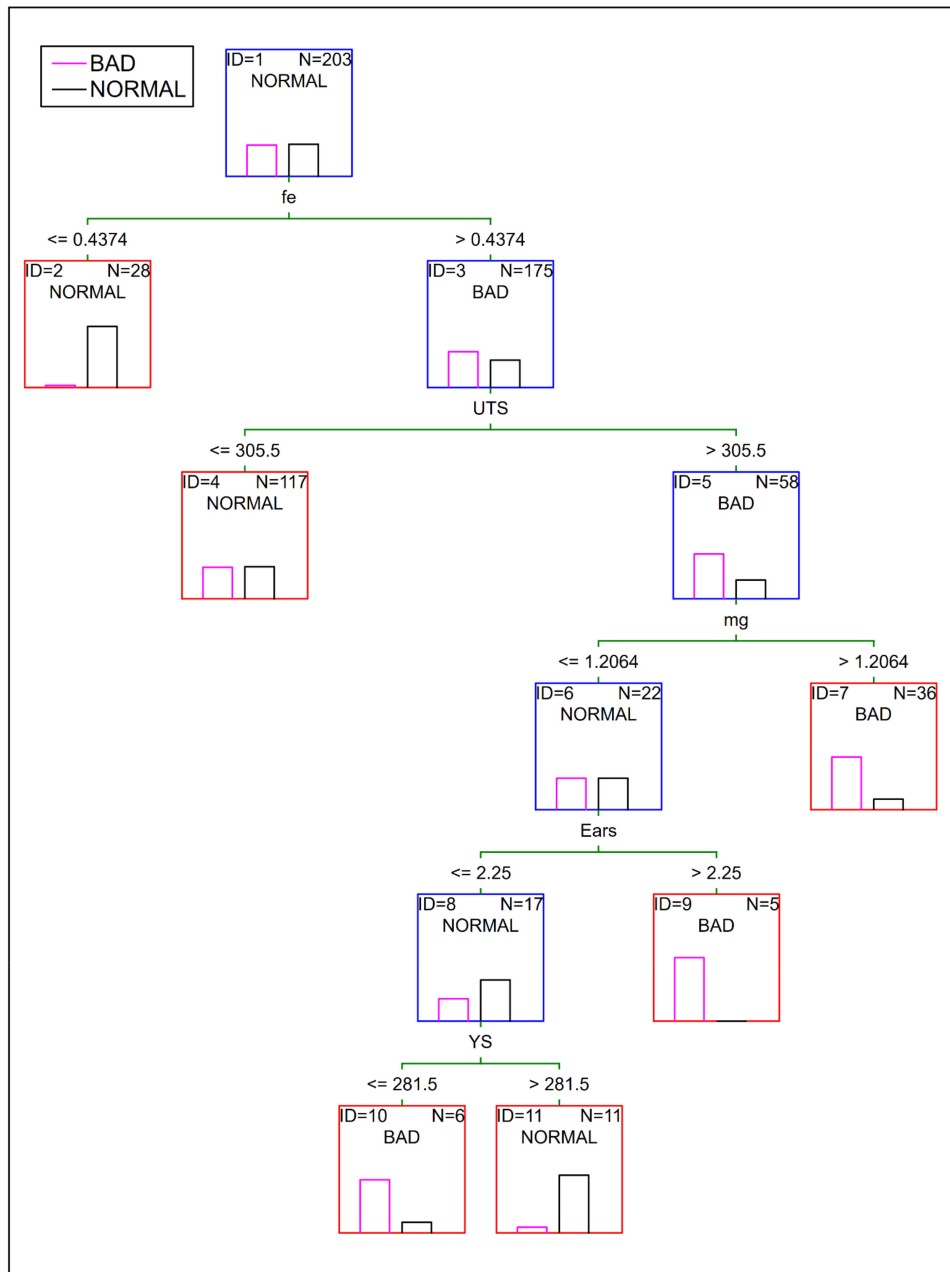

**Figure 10.** Interactive classification tree for the amount of short cans (2Q), C&RT Model.

The classification matrix (Figure 11) shows the number of cases (N.obs.) that have been correctly classified by the program for both NORMAL and BAD classes. Analyzing the classification matrix, it can be noticed that while the tree correctly recognizes cases in the NORMAL class, the error in the BAD class is very large, which means that the model has not learned to detect errors. To increase the strength of erroneous cases, another model was created in which cases only from the upper quartile of the number of damaged cans are considered to be erroneous.

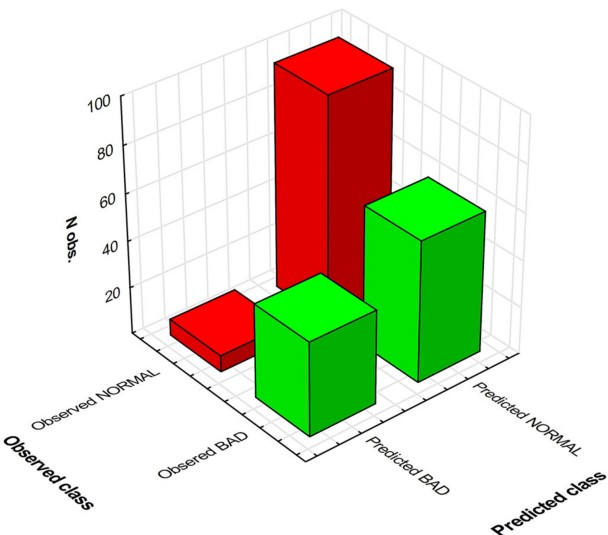

**Figure 11.** Classification matrix for the classification tree (2Q).

Figure 12 below shows the same kind of classification tree but the amount of normal coils is contained in three quartiles and called 3Q. It means that 83 defects/million produced cans and more was considered as BAD. Less than 83 defects/million was considered as GOOD. This tree shows as BAD only the worst coils (upper quartile) which can pose the greatest risk to production.

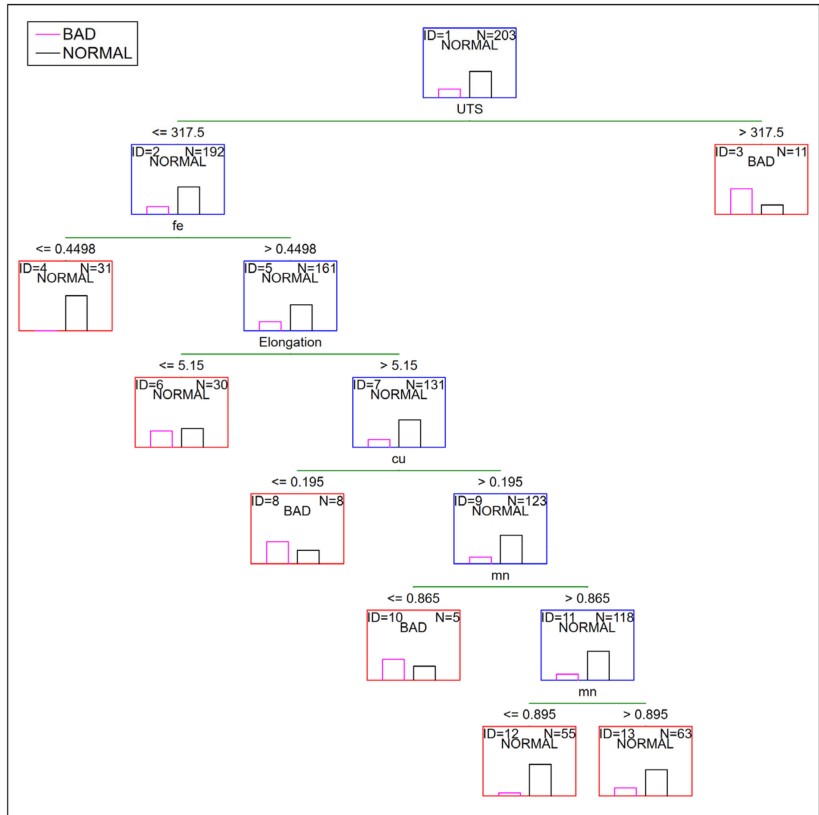

**Figure 12.** Interactive classification tree for the amount of short cans (3Q), C&RT Model.

When analyzing the classification matrix (Figure 13), it can be seen that the model is still not satisfactory in the BAD case class, although the global model error is much

smaller. For this reason, all three models were used, selecting certain rules from them (those with the highest confidence), creating a set of rules used in the hybrid model of inference (Figure 14).

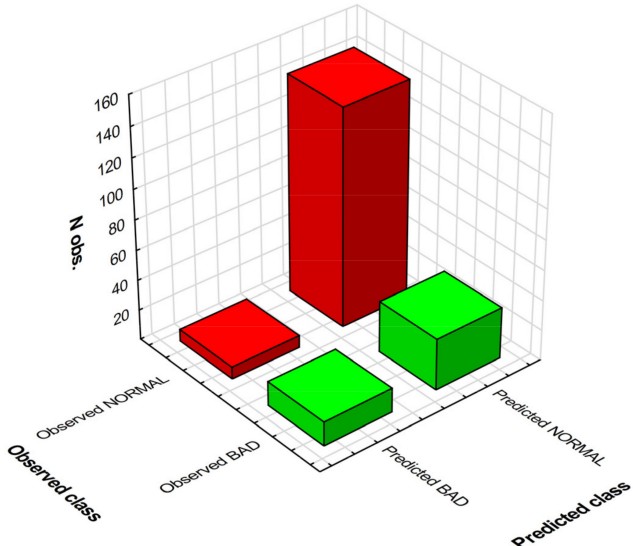

**Figure 13.** Classification matrix for the classification tree (3Q).

| Enter coil parameters | | Qualification of coils for production - answers of the statistical system | | | |
|---|---|---|---|---|---|
| YS | 280.000 | **Answer (Priority 1)** | | | |
| UTS | 305.000 | The predicted number of damaged cans/million is | 62 | Standard deviation with 3 sigma regulation (+/-) | 134 |
| Elongation | 5.700 | **Answer (Priority 2)** | | | |
| Ears | 2.083 | The predicted number of damaged cans/million below 50.4 with trustability | 0 | and rule support | 0 |
| Si | 0.220 | The predicted number of damaged cans/million above 50.4 with trustability | 0 | and rule support | 0 |
| Fe | 0.490 | **Answer (Priority 3)** | | | |
| Cu | 0.200 | The predicted number of damaged cans/million below 82.6 with trustability | 76.2 | and rule support | 31 |
| Mn | 0.910 | The predicted number of damaged cans/million above 82.6 with trustability | 0 | and rule support | 0 |
| Mg | 1.200 | | | | |

**Figure 14.** Tool for predicting the number of short cans for new coils.

Finally, rules were created for each leaf of each tree, based on the division into areas. It allows for forming a tool for predicting the amount of short cans for new coils. After entering the mechanical parameters values and chemical composition content, the program returns the result based on rules coming from three trees (Figure 14).

This tool consists of two parts. On the left side of the table in the Figure 14, the mechanical parameters of the material and the chemical composition are entered. On the right side of the table, the program generates three priorities of responses, which are the results of reasoning on the basis of the ruleset achieved from the models. From the most important to the least important. The highest priority response is taken into account in the first place during coil approval for production.

*Assessment of the Inference Mechanism*

To verify the correct operation of the program for predicting the number of damaged cans, the monthly production was taken into consideration, and 20 random sheet coils were selected from it, which were then monitored. The number of damaged cans predicted by the program was compared with the actual number of damages caused during the production (Table 6).

**Table 6.** Comparison of the predicted number of damaged cans with the actual value.

| Coil nr. | Damaged Cans/Million (Base on Production Reports) | Program Trustability 1–3 | Priority 1 | | | | Priority 2 | | Priority 3 | | | |
|---|---|---|---|---|---|---|---|---|---|---|---|---|
| | | | Predicted Number of Damaged Cans/Mln | Standard Deviation | Predicted Number of Damaged Cans below 50.4 with Trustability | and Rule Support | Predicted Number of Damaged Cans above 50.4 with Trustability | and Rule Support | Predicted Number of Damaged Cans below 82.6 with Trustability | and Rule Support | Predicted Number of Damaged Cans above 82.6 with Trustability | and Rule Support |
| 1 | 142 | 1 | 70 | 192 | 0 | 0 | 100 | 2.4 | 0 | 0 | 62.5 | 3.9 |
| 2 | 115 | 1 | 175 | 579 | 0 | 0 | 100 | 2.4 | 0 | 0 | 72.7 | 5.4 |
| 3 | 108 | 1 | 70 | 192 | 0 | 0 | 83.3 | 17.7 | 0 | 0 | 72.7 | 5.4 |
| 4 | 91 | 1 | 175.0 | 579.0 | 0.0 | 0.0 | 83.3 | 17.7 | 0.0 | 0.0 | 72.7 | 5.4 |
| 5 | 78 | 1 | 70.0 | 192.2 | 0.0 | 0.0 | 100.0 | 2.4 | 0.0 | 0.0 | 62.5 | 3.9 |
| 6 | 59 | 3 | 19.0 | 60.2 | 96.4 | 13.8 | 0.0 | 0.0 | 100.0 | 15.2 | 0.0 | 0.0 |
| 7 | 48 | 2 | 175.0 | 579.0 | 96.4 | 13.8 | 0.0 | 0.0 | 0.0 | 0.0 | 72.7 | 5.4 |
| 8 | 46 | 2 | 19.0 | 60.2 | 96.4 | 13.8 | 0.0 | 0.0 | 0.0 | 0.0 | 72.7 | 5.4 |
| 9 | 46 | 1 | 19.0 | 60.2 | 96.4 | 13.8 | 0.0 | 0.0 | 100.0 | 15.2 | 0.0 | 0.0 |
| 10 | 41 | 1 | 19.0 | 60.2 | 96.4 | 13.8 | 0.0 | 0.0 | 100.0 | 15.2 | 0.0 | 0.0 |
| 11 | 39 | 2 | 19.0 | 60.2 | 96.4 | 13.8 | 0.0 | 0.0 | 0.0 | 0.0 | 72.7 | 5.4 |
| 12 | 39 | 2 | 175.0 | 579.0 | 96.4 | 13.8 | 0.0 | 0.0 | 0.0 | 0.0 | 72.7 | 5.4 |
| 13 | 38 | 1 | 19.0 | 60.2 | 96.4 | 13.8 | 0.0 | 0.0 | 100.0 | 15.2 | 0.0 | 0.0 |
| 14 | 29 | 1 | 19.0 | 60.2 | 96.4 | 13.8 | 0.0 | 0.0 | 100.0 | 15.2 | 0.0 | 0.0 |
| 15 | 27 | 1 | 19.0 | 60.2 | 96.4 | 13.8 | 0.0 | 0.0 | 100.0 | 15.2 | 0.0 | 0.0 |
| 16 | 25 | 2 | 175.0 | 579.0 | 96.4 | 13.8 | 0.0 | 0.0 | 0.0 | 0.0 | 72.7 | 5.4 |
| 17 | 24 | 1 | 19.0 | 60.2 | 96.4 | 13.8 | 0.0 | 0.0 | 100.0 | 15.2 | 0.0 | 0.0 |
| 18 | 19 | 1 | 19.0 | 60.2 | 96.4 | 13.8 | 0.0 | 0.0 | 100.0 | 15.2 | 0.0 | 0.0 |
| 19 | 16 | 1 | 19.0 | 60.2 | 96.4 | 13.8 | 0.0 | 0.0 | 100.0 | 15.2 | 0.0 | 0.0 |
| 20 | 13 | 1 | 19.0 | 60.2 | 96.4 | 13.8 | 0.0 | 0.0 | 100.0 | 15.2 | 0.0 | 0.0 |

To distinguish between the results, a three-level "Program trustability" scale was established, where the best prediction is marked as 1 and the worst prediction is marked as 3. For a value of 1, the program results for each of the three responses coincide with the actual number of damaged cans. Value 2 means compliance of priority 1 and 2, while value 3 means compliance of priority 1 and 3, but non-compliance of priority 2, which is more important than 3. The comparison shows that 14 out of 20 examined circles show 100% compliance between the production and the prediction of the program, which is 70% of the results. A total of 5 coils are in agreement for priority 1 and 2, which is 25% of the results, and one coil for priorities 1 and 3, which is 5% of the results. Based on the verification, it can be concluded that the program predicts the number of damaged cans correctly in most cases, 25% of cases are also well-predicted for responses with priority 1 and 2. Only one case could be misleading when making the decision to release the coil for production, which may result from an insufficiently big database and, consequently, a low precision of the statistical calculations. This would suggest extending the database in the future to obtain more precise results.

## 4. Discussion

Table 3 shows that yield strength and elongation have the biggest impact on the number of damaged cans. With the increase in YS, the number of damaged cans increases. Along with the decrease in the elongation parameter, the number of damaged cans increases. With the increase in YS and UTS, the number of damaged cans increases, which confirms that the greater strength of the material reduces its plasticity and the sheet breaks earlier when forming the can. The relative elongation, on the other hand, has an inversely proportional effect on the formation, which is confirmed by the minus sign in front of the correlation value of the "Elongation" parameter.

These results are consistent with the concept that exists in material engineering. Indeed, hardening of a material, for example, after cold deformation, (increase YS) is usually accompanied by a decrease in ductility (Elongation).

Table 4 clearly shows that the increased amount of silicon (Si) is increasing the number of short cans. Similarly, increase in the amount of iron (Fe) generates a bigger scrap rate. Additionally, manganese has an indirect impact on the number of short cans by influencing the YS and UTS with the biggest correlation at the level of 0.493 and very statistically significant with $p < 0.001$.

A. Rękas at al. (2015) [9] proved that the increased plasticity margin (YS/UTS) and the strain-hardening factor causes an increasing number of defects. Another conclusion was that decreasing elongation increases the amount of damaged product. Their article shows that YS/UTS and elongation have a significant impact on spoilage but does not provide an answer as to which parameters have the biggest impact on generating a spoilage. Thanks to using a statistical method, it was possible to find a correlation and those parameters which are crucial for the process. Additionally, thanks to using the interactive regression and classification tree, it was possible to create a tool which can predict the number of short cans for new coils. The idea of an implemented decision support tool was to apply knowledge about material, actually owned by the manufacturer without additional tests. There are also parameters which could have potentially influenced the defect generation, such as tool wear which was analyzed by A. Rękas at al. [9] anisotropy, orientation distribution function and crystallographic texture analyzed by Andre Luis Teixeira Martins at al. [14] and other parameters such as strain-hardening coefficient, roughness, etc., which can be a good extension of this research in the future.

## 5. Conclusions

The paper presents the possibilities of applying IT tools, especially decision tree induction (C&RT) used for the creation of an approximation model of the prediction of industrial processes of deep drawing of cylindrical thin-walled products from aluminum sheets. With the current configuration of the process parameters measurements and input

data, i.e., batch parameters, the statistical analysis showed that the relationships among the parameters are not significant, and the measured correlations do not allow for the development of satisfactory regression models. It is an example of forecasting in the reality of uncertain data (we rely, among others, on data provided by various suppliers with unknown measurement conditions) and incomplete (a limited number of parameters). A new point of view in the forming process is to modify the technological process by changing the input material when too many defects are expected. Thanks to the development of a tool for predicting the number of defects, it is possible to manage the production process in such a way that during periods of increased production, coils that pose too high a risk of a large amount of defective product can be postponed and used in low season for beverage cans. The aim of the work was not to modify the forming process itself, but to develop a production forecasting tool. The authors, striving for a possible approximation of the quality parameter, which was the number of "short cans", used three decision tree models, combining into one base the rules with high confidence indices and possibly a small variance in leaves. This method consisted of developing a regression model, and then two classification models, on the discretized dependent variable cut off at 2 and 3 quartiles, respectively. This made it possible to develop a model of inference with three degrees of precision, and then to integrate the result of inference in the form of an ensemble model, where the decision is given based on the voting of each degree. The obtained result is still not errorless, but in circumstances where there is a small training dataset so far, it turned out to be satisfactory. In the long run, when we have larger training datasets, where the ranges of variability will be covered to a greater extent, these models will allow us to capture dependencies in a much more precise scope, and thus, inference will enable generally consistent and unambiguously quantitative results.

The obtained results of the developed inference model enable a rough estimation of the values of the short-can index on the basis of the relations between chemical components and the index of short cans, but also of an indirect correlation in the influence of chemical elements on mechanical properties, such as yield strength, ultimate tensile strength and elongation.

**Author Contributions:** Conceptualization, W.B. and A.M.; methodology, W.B., K.R. and A.M.; software, W.B. and K.R.; validation, W.B., K.R. and A.M.; investigation, W.B., K.R. and A.M.; writing—original draft preparation, W.B. and K.R.; writing—review and editing, W.B., K.R. and A.M.; visualization, W.B. All authors have read and agreed to the published version of the manuscript.

**Funding:** The investigation was not sponsored.

**Institutional Review Board Statement:** Not applicable.

**Informed Consent Statement:** Not applicable.

**Data Availability Statement:** Data are provided within the article.

**Conflicts of Interest:** The authors declare no conflict of interest.

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
