# Peer review of "Influence of Materials Parameters of the Coil Sheet on the Formation of Defects during the Manufacture of Deep-Drawn Cups"

_processes, doi:10.3390/pr10030578_

Round 1

Reviewer 1 Report

Manuscript ID processes-1631241;

Type: Article;

Title: "Influence of materials parameters of the coil sheet on the formation of defects during the manufacture of deep-drawn cups";

By Authors:  Wojciech Baran, Krzysztof Regulski, Andrij Milenin.

The article presented for review is well written.
However, it requires a minor correction before being published in the Journal Processes.  

  1. The caption of Figure 1 is imprecise,
        "Figure 1. Analyzed defect -" short can "",
         there is no description of (a) and (b).
         Perhaps it is worth highlighting  information about the defects on the figures.
  2. Why in the tables:
    "Table 1. Mechanical parameters range for aluminum alloy 3104 [13]",
    "Table 2. Chemical composition of aluminum alloy 3104 [14]",
    literature data are given.
    Was it not possible to define these parameters on the basis of the author's own research?
  3. Poor quality Fig.2: "Figure 2. Aluminum beverage can technology process". It is worth correcting the descriptions.
  4. Tables: "Table 3. Correlation between the mechanical parameters and the index of short cans for all coils", "Table 4. Correlation between the chemical components and the index of short cans, YS, UTS, and 239
    Elongation",  are not very legible, maybe it is worth improving the graphic form?
  5. I think "Figure 13. Tool for predicting the number of short cans for new coils" should be converted to Table.

Author Response

Dear reviewer,

all ma answers are in attached file.

Reviewer 2 Report

  1. Please indicate the unit of chemical composition in tables, figures, and sentences.
  2. Please clarify if tensile tests were carried out for the coils after final cold rolling, or previous processing steps.  
  3. Please clarify the direction of tensile tests (Rolling direction?)
  4. Please discuss the other properties of the coils in terms of the relevance to the defects. Anisotropy of mechanical properties, R value, n value, texture, etc might have something to do with the defect occurrence.  

Author Response

Dear reviewer,

all my comments are in attached file.
